# Associations Between Non-Steroidal and Steroidal Anti-Inflammatory Drug Use, Welfare, and Milk Production in Dairy Sheep: A Multivariate Study

**DOI:** 10.3390/ani15081104

**Published:** 2025-04-11

**Authors:** Nikolaos Tsekouras, Mathis A. B. Christodoulopoulos, Eleftherios Meletis, Christos Kousoulis, Polychronis Kostoulas, Vasileios Pantazis, Vasileios G. Papatsiros, Konstantina Dimoveli, Dimitrios Gougoulis

**Affiliations:** 1Clinic of Medicine, Faculty of Veterinary Science, University of Thessaly, 43100 Karditsa, Greece; nitsekou@uth.gr (N.T.); vpapatsiros@uth.gr (V.G.P.); kdimoveli@uth.gr (K.D.); 2Faculty of Sciences, Aix-Marseille University, 13007 Marseille, France; mathis.christodoulopoulos@etu.univ-amu.fr; 3Faculty of Public and Integrated Health, University of Thessaly, 43100 Karditsa, Greece; emeletis@outlook.com (E.M.); pkost@uth.gr (P.K.); 4Agricultural Cooperative of Cow and Sheep Farmers of Western Thessaly, Lazarina, 43060 Karditsa, Greece; kouschristos@uth.gr; 5Biochemistry and Biotechnology Department, University of Thessaly, 41500 Larissa, Greece; vapantazis@uth.gr

**Keywords:** non-steroidal anti-inflammatory drugs (NSAIDs), corticosteroids (SAIDs), flunixin meglumine, meloxicam, milk yield, pain management, welfare assessment, sheep

## Abstract

This study investigates the use of pain relief medications by sheep farmers and reveals that many do not fully utilize the available options. The findings suggest that the limited adoption of analgesic drugs in sheep farming may stem from insufficient information and a lack of appropriate guidelines in the region. Additionally, the results indicate that larger farms are more likely to employ pain relief treatments, potentially contributing to improved productivity. These findings highlight the need for better education and resources to support sheep farmers in implementing effective pain management practices. By addressing these gaps, this study aims to promote enhanced animal welfare and overall farm performance. Future research is needed to build on these findings and further explore the implications of analgesic use in sheep farming.

## 1. Introduction

Sheep are among the most important livestock species in Europe, ranking third in population, with a particularly strong presence in Mediterranean countries such as Greece [1]. The Greek sheep farming industry comprises approximately 7.3 million sheep across 83,000 farms, with a predominant focus on dairy production [2]. The milk produced is primarily used for feta cheese, a key export product under the Protected Designation of Origin status. However, over the past decade, the number of small- and medium-sized farms has declined, leading to an increase in larger operations with over 500 ewes, relying on intensive housing systems and imported concentrate feeds [3,4].

Although farm animal welfare has been the focus of research for decades, much attention has been directed toward intensively farmed species, with less emphasis on sheep. The shift toward more intensive farming systems has raised concerns about animal welfare, as these systems often restrict natural behaviors, leading to stress and potential health issues [5]. Consumers increasingly prioritize high welfare farming practices, placing pressure on the industry to mitigate pain and improve overall conditions for livestock [6,7].

Pain in sheep may stem from various sources, including diseases such as mastitis and lameness, routine management practices such as tail docking, and environmental factors such as inadequate housing conditions [8]. As prey animals, sheep tend to mask signs of pain, making it challenging for farmers and veterinarians to assess and manage discomfort effectively [9,10,11,12]. If left untreated, pain can lead to behavioral changes, reduced feed intake, and decreased productivity [13,14,15,16]. Under current EU legislation, farm animals are considered sentient beings [17] and are therefore recognized to suffer and feel pain. Freedom from pain is a fundamental component of animal welfare [18].

Providing analgesia to food-producing animals presents several challenges, including a limited number of approved medications, regulatory restrictions, frequent need for repeated administration, high costs, and meat and milk withholding periods [19]. Many analgesics also have a delayed onset and short duration of action, requiring multiple doses to ensure effective pain relief. This poses difficulties for producers and veterinarians, as animals may continue to experience pain for several days after a painful procedure, necessitating careful management and repeated handling [20]. Compared to other farm animal species, the use of non-steroidal anti-inflammatory drugs (NSAIDs) in sheep farming remains relatively low. Research conducted in Northern Ireland indicated that 27% of sheep farmers do not utilize any NSAIDs [21]. The primary reasons for this low adoption include the following: producers and veterinarians citing the costs of treatment, the absence of analgesic drugs approved for use in sheep, concerns about drug residues in tissues or milk, and insufficient scientific data assessing pain responses before and after treatment [22,23].

Currently, the most effective options for on-farm analgesia include local anesthetics and NSAIDs such as flunixin or meloxicam [19]. NSAIDs exert their effects by inhibiting cyclooxygenase (COX) enzymes, which are involved in the biosynthesis of prostaglandins (PGs) and thromboxane (TX). This inhibition reduces prostaglandin production, which is a key mediator of pain and inflammation [24].

By minimizing inflammation, NSAIDs not only improve animal comfort but also accelerate the healing process, reducing recovery time from injuries and disease. Addition-ally, by limiting inflammatory damage, they decrease the likelihood of permanent tissue impairment, allowing animals to regain their pre-inflammation levels of productivity [8]. The use of NSAIDs has been linked to improved production outcomes, but their precise effects on dairy sheep welfare and milk yield remain insufficiently studied. Similarly, corticosteroids, also known as steroidal anti-inflammatory drugs (SAIDs), such as dexamethasone, are used for their anti-inflammatory effects. However, their role in pain relief and their long-term implications for animal health require further investigation [25,26,27]. Ad-dressing pain in livestock is critical not only from an ethical perspective but also from an economic one, as poor welfare conditions can negatively impact production efficiency and product quality [28]. The concept of positive welfare holds significance not only in the scientific realm but also in society, particularly among farmers and the general public, who consider both positive and negative welfare aspects when evaluating livestock production systems [29].

In Greece, the lack of NSAID approvals for sheep has led to the off-label use of medications licensed for cattle, posing challenges in dosage standardization and efficacy assessment. The NSAIDs most commonly used by veterinary professionals are flunixin meglumine (FM), meloxicam (MX), and ketoprofen (KP). FM inhibits both COX-1 and COX-2, while MX is known for its selective inhibition of COX-2, specifically targeting inflammatory processes [30]. Although both KP and FM exhibit significant COX-1 activity, the side effects associated with KP are typically minimal due to its shorter circulation time compared to FM [31].

Given the growing emphasis on “welfare-focused livestock management”, it is crucial to evaluate the impact of NSAIDs and corticosteroids on both welfare parameters and milk production in modern sheep farming systems. While increased NSAID use may indicate the presence of health challenges within a farm, it can also reflect a proactive management approach that prioritizes effective pain relief and disease control. Additionally, welfare and NSAID use share a bidirectional relationship: poor welfare conditions may necessitate higher NSAID use, but proper analgesic administration can, in turn, enhance welfare and prevent long-term tissue damage, allowing animals to return to optimal productivity.

This study investigates the relationship between NSAID and corticosteroid use, welfare indicators, and milk yield in Greek dairy sheep farms. By assessing these factors in conjunction with farm management practices, it aims to clarify the role of NSAIDs in balancing animal welfare and productivity. The findings will contribute to evidence-based recommendations for optimizing sheep farm management strategies and promoting better analgesic use in the sector.

## 2. Materials and Methods

### 2.1. Study Area, Sampling, and Data Collection

This study was conducted between 1 September 2022 and 31 August 2023, covering a full lactation period, in the region of West Thessaly. Thirty randomly selected dairy sheep farms, located near Karditsa (43100) and Trikala (42100), participated in this study. The total population of animals was 5250, with individual farm sizes ranging from 45 to 700 animals. The distribution of selected farms is presented in Figure 1. All animal procedures regarding animal care and handling were approved by the Institutional Ethical Committee (University of Thessaly, approval number: 102, date of the approval: 16 November 2021). This observational study involved minimal animal handling. All procedures related to animal care and observation were approved by the Institutional Ethical Committee of the University of Thessaly (approval number: 102, date of approval: 16 November 2021). Written informed consent was obtained from all participating farmers for data collection and the inclusion of their farms in this study.

The selection of 30 farms was based on both practical constraints (e.g., logistical and financial limitations) and statistical considerations to ensure a representative sample. Conducting detailed welfare assessments, collecting management data, and monitoring medication usage over an entire milking period posed significant challenges for a larger sample size. A power analysis indicated that with n = 30, this study had sufficient power to detect moderate-to-large effect sizes (Cohen’s d ≥ 0.7) at a 95% confidence level.

The sample was randomly selected using simple random sampling from a list of registered dairy sheep farms in the region, ensuring equal probability of selection among farms with similar characteristics. Farms were eligible for inclusion if they adhered to routine vaccination protocols for clostridial diseases and brucellosis and operated under an intensive farming system (i.e., indoor housing and reliance on formulated feed). All the animals belonged to indigenous breeds crossed with Lacaune sheep. Specifically, all selected farms reared crossbred Lacaune sheep under intensive housing systems without access to grazing. The selected farms ranged in size (from 45 to 700 animals) but shared common management practices, providing a representative dataset of intensive dairy sheep farming in West Thessaly.

The animals’ diet was based on a mixture of grains based on corn. Additionally, alfalfa hay was provided throughout the year, in varied quantities, according to the reproductive stage of each animal group and hay ad libitum all year around. The veterinary services provided on the farms, apart from vaccinations, were the supervision of the farm bimonthly, ultrasound examinations (45–60 days after mating), and any other urgent procedures such as caesarean sections, etc.

However, the selection process did not explicitly control for potential confounding factors such as specific feeding regimens, minor genetic variations, or microclimatic differences, which may influence both farm management strategies and animal welfare indicators. Furthermore, as this study was conducted exclusively in West Thessaly, the findings may not be directly generalizable to other regions with different climatic conditions and farming systems.

Farm visits were conducted monthly. During each visit, data on milk yield, milk composition (fat and protein), NSAID use (FM and MX), and corticosteroid (DEX) administration were recorded based on veterinarian-issued invoices. NSAID usage was standardized according to the number of vials purchased: for meloxicam (MX), 30 mL vials with a concentration of 20 mg/mL were used, with a recommended dosage of 2.5 mL per 100 kg of body weight for three days; for flunixin meglumine (FM), 100 mL vials with a concentration of 8.5% were used, with a recommended dosage of 2 mL per 45 kg of body weight for three days. Dexamethasone (DEX) usage was recorded based on 50 mL vials with a concentration of 2 mg/mL, with a recommended dosage of 1 mL per 25 kg of body weight for three days. For the classification of farms based on pain management strategies, four distinct groups were established: (1) farms administering only NSAIDs (flunixin meglumine, meloxicam or ketoprofen), (2) farms using only dexamethasone, (3) farms employing a combination of NSAIDs and corticosteroids (mixed use), and (4) farms that did not administer any pharmacological treatment. Given the similarities in their anti-inflammatory and analgesic mechanisms, flunixin meglumine and meloxicam were analyzed as a single category under NSAID use. The Body Condition Score (BCS) was assessed by trained observers using a standardized 1–5 scale [32].

In the first farm visit, we recorded the number of animals on each farm, the farmers’ attainment of formal higher education (yes/no), the farmers’ age categorized as <35 years, 35–50 years, and >50 years, the number of animals per worker, the use or non-use of estrus synchronization protocols, and the housing conditions. Housing conditions were classified according to EU Regulation 2092/91 [33] and expert assessment, categorized into three levels: “Suboptimal” (facilities built before 2000 with inadequate space; <1.5 m^2^/ewe and lamb), “Moderate” (adequate space but older infrastructure), and “Optimal” (modern facilities with sufficient space).

Welfare assessment was conducted by an experienced veterinary researcher using a modified scoring system based on Poulopoulou et al. [34], originally designed for cattle welfare and adapted for sheep (Table 1). The evaluation was performed prior to the assessment of anti-inflammatory drug administration, ensuring that the process was blinded to NSAID and SAID usage. Farms were categorized by the score they achieved into four categories: Category “A” for farms scoring 80.1 to 100 points, “B” for farms scoring 60.1 to 80 points, “C” for farms scoring 40.1 to 60 points, and “D” for farms scoring up to 40 points according to the criteria in Table 1. The scoring framework was aligned with regulations from the Greek Ministry of Rural Development and Food [35] and Zygogiannis [36].

### 2.2. Statistical Analysis

The dataset includes information from 30 farms, selected to balance practical constraints with the need for a representative sample. Given the relatively small sample size, multiple statistical approaches were implemented to enhance the robustness of the findings. Propensity Score Matching (PSM) was applied to reduce confounding effects by creating comparable groups of farms with and without NSAID use. Additionally, bootstrap-ping methods were used to estimate parameter variability and generate more reliable confidence intervals.

To assess the impact of NSAIDs on milk production and welfare, a Multivariate Analysis of Variance (MANOVA) was conducted in SPSS, ensuring the validity of the results through multivariate normality checks. Logistic regression models were employed in both R and SPSS to examine factors associated with NSAID administration and the likelihood of achieving a high welfare score (category A), with independent variables including farm size, housing conditions, and estrous synchronization.

Further methodological controls were implemented to strengthen causal inference. Selection bias in PSM was assessed by comparing pre- and post-matching covariate distributions in R. Additionally, multicollinearity among independent variables was evaluated to prevent model distortion. Univariate analysis identified potential predictors (*p* ≤ 0.25), which were refined through backward elimination, retaining only statistically significant variables (*p* < 0.05) in the final models. The goodness-of-fit of logistic regression models was assessed using the Pearson chi-square statistic, confirming model adequacy. All statistical analyses were conducted using R (version 4.4.1) and SPSS [version 29.0.0.0 (241)] [37].

## 3. Results

### Descriptive Statistics

The descriptive analysis of the study variables revealed substantial variation in herd size, milk yield, and key welfare indicators among the 30 dairy sheep farms. Total milk yield per farm ranged from 5394 kg to 236,739 kg, with a mean production of 50,589 kg (SD = 49,946 kg). Milk yield per sheep also exhibited variability, ranging from 119.87 kg to 383.81 kg, with an average of 261.26 kg (SD = 67.63 kg). Regarding milk composition, fat percentage varied between 5.51% and 7.01%, with a mean of 6.12% (SD = 0.38%), while protein content ranged from 4.67% to 6.18%, with a mean of 5.61% (SD = 0.30%).

The classification of farms based on pain management strategies showed that 26.7% (n = 8) did not administer NSAIDs or corticosteroids, while 73.3% (n = 22) implemented some form of pharmacological pain management. Specifically, 30.0% (n = 9) of farms used only NSAIDs (flunixin meglumine or meloxicam), 16.7% (n = 5) relied exclusively on corticosteroids, and 26.7% (n = 8) adopted a mixed approach, combining both NSAIDs and corticosteroids (Figure 2). Overall, NSAIDs were administered in 56.7% of farms and corticosteroids in 43.3%, while combined NSAID and corticosteroid use was less frequent (26.7%). The distribution of treatment categories was skewed, with most farms adhering to a single treatment strategy rather than employing a combined approach.

Farm management characteristics also exhibited considerable variation. The number of animals per worker ranged from 30 to 200, with a mean of 88.06 (SD = 39.28), reflecting substantial differences in labor intensity across farms. Detailed results are presented in Table 2.

The median Body Condition Score (BCS) was 2.00 (IQR: 2.00–3.00), indicating that most animals were within an acceptable body condition range. Housing conditions had a median score of 2.00 (IQR: 2.00–3.00), suggesting that the majority of farms fell into the moderate category in terms of infrastructure and space allocation. Avoidance behavior had a median score of 0.00 (IQR: 0.00–10.00), demonstrating a wide range of responses to human presence. Hoof conformation showed a median of 2.50 (IQR: 0.00–10.00), reflecting variability in hoof care and trimming practices across farms. Lameness prevalence had a median of 5.00 (IQR: 0.00–5.00), while dystocia occurrence had a median of 10.00 (IQR: 5.00–10.00), highlighting differences in reproductive health among the studied farms.

The Total Welfare Score ranged from 45 to 95, with a median of 70.00 (IQR: 55.00–86.25), reflecting the variation in welfare conditions among the sampled farms (Figure 3). More detailed information on the distribution of these variables is presented in Table 3.

The analysis of categorical variables revealed that estrous synchronization protocols were implemented in 22 out of 30 farms (73.3%), while 8 farms (26.7%) did not use them. Regarding education level, 22 farmers (73.3%) had only basic education, whereas 8 (26.7%) had completed tertiary education.

The ANOVA results indicated a statistically significant difference in milk yield per sheep (kg) among the treatment groups [F (3,26) = 3.102, *p* = 0.044]. However, no significant differences were observed for mean fat percentage [F (3,26) = 1.418, *p* = 0.260] or mean protein percentage [F (3,26) = 1.407, *p* = 0.263]. Post-hoc Bonferroni comparisons did not reveal statistically significant differences between specific treatment groups for milk yield per sheep (all *p*-values > 0.05). Similarly, no significant differences were detected between groups for fat or protein content.

A multiple linear regression model was employed to assess the effects of treatment use, farmer characteristics, and management practices on total milk yield (kg). The model demonstrated strong predictive power, explaining 80% of the variance in total milk yield (R^2^ = 0.800). The ANOVA for the regression model indicated statistical significance [F (7,22) = 12.556, *p* < 0.001], suggesting that at least one of the independent variables significantly influenced total milk yield.

Examination of regression coefficients (Table 4) revealed that workers per animal was a strong positive predictor (B = 610.027, *p* < 0.001), highlighting the importance of labor availability in milk production efficiency. Farmer’s education (university degree) was significantly associated with higher total milk yield (B = 37,156.574, *p* = 0.006), suggesting that more educated farmers implemented improved management strategies. Among pharmacological treatments, flunixin use had the most pronounced positive effect (B = 42,616.334, *p* = 0.006), followed by meloxicam use (B = 28,623.026, *p* = 0.047), supporting the hypothesis that NSAID administration contributed to enhanced milk yield. Conversely, the effects of mixed use (NSAIDs + corticosteroids) (B = −30,338.408, *p* = 0.171), corticosteroid use (B = −3386.599, *p* = 0.848), and farmer’s age (B = −7684.890, *p* = 0.346) did not reach statistical significance, indicating that these factors did not have a consistent impact on total milk yield.

Collinearity diagnostics confirmed no major multicollinearity concerns among the predictors, with variance inflation factor (VIF) values ranging from 1.326 to 4.098, ensuring the statistical reliability of the model. Figure 4 presents the estimated regression coefficients (B) for the significant predictors of total milk yield, illustrating the positive associations of education status, flunixin use, and meloxicam use with milk yield.

Table 5 summarizes the results of the five logistic regression models implemented in this analysis. Each model is described by its intercept, which represents the baseline log-odds of the dependent variable being 1 (i.e., the occurrence of the event, such as NSAID administration) when all independent variables are set to zero. The point estimate reflects the change in the log-odds of the dependent variable occurring for a one-unit increase in the independent variable. Model 1 examined the association between flunixin meglumine administration and the independent variables. A positive but small association was observed between herd size and flunixin meglumine use, with an odds ratio of 1.01 (95% CI: 1.005–1.025, *p* < 0.01), indicating that larger herds were more likely to administer this NSAID. Conversely, meloxicam administration was negatively associated with herd size, with an odds ratio of 0.95 (95% CI: 0.88–0.99, *p* = 0.03), suggesting that smaller herds were more inclined to use meloxicam. However, total milk yield was positively correlated with meloxicam use, with an odds ratio of 1.002 (95% CI: 1.001–1.004, *p* = 0.025), indicating that herds with higher milk production were more likely to administer meloxicam. Regarding dexamethasone administration, farmers with higher education (university degree) were significantly more likely to use this corticosteroid, with an odds ratio of 6.43 (95% CI: 1.15–52.25, *p* = 0.04). Finally, NSAID administration overall was positively associated with total milk yield, with an odds ratio of 1.05 (95% CI: 1.018–1.11, *p* = 0.02), suggesting that farms with greater milk production were more likely to implement NSAID treatments.

Higher total milk yield was positively associated with improved welfare scores, with an odds ratio of 1.04 (95% CI: 1.02–1.08, *p* = 0.004), indicating that farms with greater milk production were more likely to achieve higher welfare scores.

## 4. Discussion

Our findings indicate that 26.7% of sheep farmers did not utilize any analgesic medications, a percentage that aligns with previous reports from Northern Ireland [21]. A former study from Greece confirms the limited use of non-steroid analgesic agents among sheep farmers in cases of lameness [38]. In contrast, NSAID and corticosteroid use in other livestock sectors appears substantially higher, with reported usage exceeding 80% in swine and 85% in cattle [39,40]. This discrepancy suggests that on-farm management practices and farmer education may not sufficiently support the adoption of NSAIDs and corticosteroids for pain management in sheep. Considering that the diagnosis and complications of a disease are regarded as significant factors affecting animal welfare [7], the limited use of these medications, especially when compared to their established role in antimicrobial protocols, highlights potential gaps in awareness and implementation. Additionally, the lack of approved NSAID indications for sheep in Greece likely contributes to the low usage, making it harder for veterinarians and farmers to integrate these treatments into routine care. The considerable proportion of non-users underscores the need for further investigation into the barriers preventing NSAID adoption. Targeted educational initiatives aimed at raising farmer awareness of the benefits of analgesic use could play a crucial role in improving pain management practices in sheep farming systems.

Milk yield is a key determinant of profitability in dairy sheep farming, with Papanikolopoulou et al. [3] identifying milk sales as the primary source of income in sheep production systems. Several factors influence milk yield, including genetics, nutrition, and management practices [41]. Notably, sheep milk has a higher solid content than cow or goat milk, making it particularly valuable for cheese production [42,43]. Our findings indicate that larger farms with a higher worker-to-animal ratio were more likely to administer NSAIDs and achieve higher milk yields per animal. Increased labor availability facilitates more effective health monitoring and timely medical interventions, potentially improving production efficiency. These results align with a previous study in Greece, which demonstrated that NSAID administration—particularly flunixin—combined with improved management practices led to enhanced milk yield and quality in cases of clinical mastitis [44].

The higher milk production observed in larger sheep farms is consistent with previous studies indicating that larger and more intensive operations achieve significantly higher incomes, both from milk and meat production [3,45,46]. These findings suggest that sheep farming is progressively adopting efficiency-driven management strategies, mirroring trends seen in dairy cattle farming, albeit with some delay. Earlier research has highlighted the scale effect in livestock production, where farm expansion and intensification contribute to improved economic outcomes [47,48].

Flunixin and meloxicam are widely recognized as the most effective NSAIDs for managing pain, inflammation, and fever in livestock, either as standalone treatments or in combination with local anesthetics [19]. Our findings indicate that herd size was positively correlated with flunixin use but negatively associated with meloxicam administration, a trend that may reflect the distinct pharmacokinetic properties of these NSAIDs. Meloxicam requires up to 12 h to reach full effectiveness but provides prolonged action for up to 48 h, whereas flunixin has a more rapid onset but a shorter duration of action (<24 h) [19]. A recent study on sheep demonstrated comparable levels of post-operative analgesia between flunixin meglumine and meloxicam [49].

Economic and logistical factors may also influence NSAID selection. Flunixin meglumine is marketed in larger 100 mL formulations in Greece, whereas meloxicam is available in smaller 30 mL vials, making the latter more practical for smaller herds. Additionally, farmer perceptions regarding brand-name recognition and product efficacy may contribute to a preference for flunixin among larger, more intensive, and profit-driven operations.

This study identified a positive correlation between improved welfare conditions—including better hygiene, optimal body condition, and higher welfare scores—and increased milk production. However, it is important to note that milk yield alone is not a definitive indicator of welfare status in cattle, as it does not necessarily capture broader aspects of animal well-being [50]. Nevertheless, poor baseline welfare can lead to reduced milk production, with lameness, for instance, being associated with milk yield losses exceeding 60 L per lactation period [51]. Similarly, suboptimal hygiene standards can negatively impact udder health, increasing the incidence of mastitis and subsequently reducing milk yield [52]. In this study, higher welfare status was associated with greater milk production, yet NSAID administration did not correlate with improved welfare scores. This finding may be attributed to poor welfare conditions increasing disease incidence, thereby leading to a greater need for analgesic intervention. While NSAIDs facilitate recovery from illness and contribute to improved animal welfare [40], our results indicate that their use alone is not a reliable welfare indicator. Although higher analgesic use was observed in farms with lower welfare scores, this does not imply that pain management is less important in high-welfare systems. Painful conditions such as lameness, dystocia, or injuries may still arise even under optimal management. Therefore, the timely and appropriate use of analgesics should be regarded as a fundamental component of good farming practice across all systems, regardless of their overall welfare status [53].

Ensuring effective animal welfare, in line with veterinary and industry standards, requires the administration of pain relief to animals in pain on all farms. While the ANOVA indicated a statistically significant association between treatment use and milk yield per sheep, post-hoc Bonferroni comparisons did not reveal significant differences between specific treatment groups. This may be related to the limited sample size, which restricts the statistical power of pairwise comparisons. Moreover, it is possible that treatment effects on milk yield, although present, are modulated by underlying management differences across farms, making group distinctions less pronounced. Therefore, these findings should be interpreted cautiously and considered in conjunction with the broader results of the multivariate regression models. Future research with larger sample sizes and controlled experimental settings could clarify these associations and determine whether NSAID administration consistently influences milk yield.

There is a notable gap in the literature regarding the relationship between welfare practices in sheep farming and the use of analgesic medications. Our study demonstrated low rates of NSAID and SAID usage among sheep farmers, highlighting potential short-comings in pain management strategies within this sector. Unlike NSAIDs, corticosteroids such as dexamethasone have been widely used for their anti-inflammatory properties in various livestock species, including for analgesia, despite limited clinical evidence sup-porting their efficacy for pain relief and concerns about potential side effects [54,55]. Our findings indicate that farmers with advanced education (university degree) were significantly more likely to administer dexamethasone, suggesting that educational background may influence medication preferences. However, a recent study in Northern Ireland re-ported meloxicam as the most commonly used analgesic in sheep (73%), followed by dexamethasone (16%), flunixin (7%), and ketoprofen (4%) [21]. Although NSAIDs are well-documented in the literature as effective pain-relieving agents that enhance animal welfare [19], our findings suggest that their adoption in sheep farming remains significantly lower than in other livestock sectors [8], such as swine and cattle, where usage rates exceed 80% [39,40]. This discrepancy may stem in part from the lack of approved NSAID indications for sheep in Greece, posing regulatory and practical challenges to their widespread adoption. While this study provides valuable insights, its relatively small sample size may limit the generalizability of the findings. Future research with a larger and more diverse sample could offer a more comprehensive understanding of analgesic use patterns associated welfare outcomes, and potential interventions to enhance pain management in sheep farming.

Moreover, while NSAIDs are primarily used for their intended purpose of pain relief and inflammation control, corticosteroids such as dexamethasone are often administered for additional therapeutic indications, including the management of pregnancy toxemia in sheep [56]. This broader range of applications introduces a potential confounding factor when assessing the role of SAIDs in farm management and welfare. Future research should consider differentiating corticosteroid use based on its intended purpose to better isolate its effects on welfare outcomes.

As societal and ethical concerns regarding animal welfare continue to grow along-side the advancement of sustainable agrifood systems, the implementation of effective on-farm welfare strategies—particularly in pain management—will be increasingly critical. While addressing this challenge is inherently complex, a multimodal approach that integrates veterinary expertise, knowledge transfer, and farm-level risk assessment is essential for improving livestock health and welfare. The veterinary profession plays a key role in raising awareness, guiding best practices, and identifying tailored solutions to enhance pain management in food-producing animals [44]. One of the primary obstacles to effective pain management is the objective assessment of pain in livestock, which remains challenging and is a key consideration in the regulatory approval of analgesic drugs across different countries. The importance of adequate analgesia is widely recognized and has been embedded in legislation in numerous regions [12]. Among the available analgesic options, NSAIDs are the only class of analgesics approved for food-producing animals in the European Union that provide long-acting pain relief (24–72 h per dose) [57]. Despite their recognized efficacy, there is a notable lack of NSAID products licensed specifically for sheep. Within the EU, NSAID use in sheep is permitted through “The Cascade System”, which allows veterinarians to prescribe medications authorized for other food-producing species (e.g., cattle) on a case-by-case basis to safeguard animal welfare [58]. While this regulatory mechanism enables pain management in sheep, it also places an increased legal and administrative burden on prescribing veterinarians. In the absence of species-specific pain relief medications, extra-label drug use remains necessary to address welfare concerns in livestock [19]. However, further research is needed to better characterize the duration of pain in sheep, develop multi-day pain management strategies, and establish cost-effective and practical dosing regimens [19]. These efforts are essential for advancing welfare standards and ensuring sustainable and ethical livestock production.

## 5. Conclusions

This study highlights the associations between NSAID and SAID use, farmer education, animal welfare, and milk production in Greek dairy sheep farms. The positive correlation between total milk yield and welfare scores underscores the importance of implementing effective welfare practices to enhance productivity. Furthermore, the significant association between farmers’ education levels and total milk yield highlights the crucial role of education in promoting improved management strategies. However, the notable finding that 26.7% of sheep farmers do not use any analgesic medications reveals a significant gap in pain management practices within the sector. While NSAID administration (e.g., flunixin, meloxicam) was linked to higher milk yields, its lack of correlation with improved welfare scores suggests that poor welfare conditions may increase disease prevalence, necessitating greater reliance on analgesics. Despite the correlation with low welfare levels, pain relief medications remain a valuable tool for all sheep farms. Our findings emphasize the urgent need for targeted education and awareness programs to promote the judicious use of analgesics and elevate overall welfare standards. Addressing these gaps will enable stakeholders to implement more sustainable management practices that improve animal welfare and productivity. Future research should focus on developing effective strategies for integrating analgesic use into routine farming operations, ensuring optimal health and welfare outcomes for sheep while enhancing the economic sustainability of dairy sheep farming.

## Figures and Tables

**Figure 1 animals-15-01104-f001:**
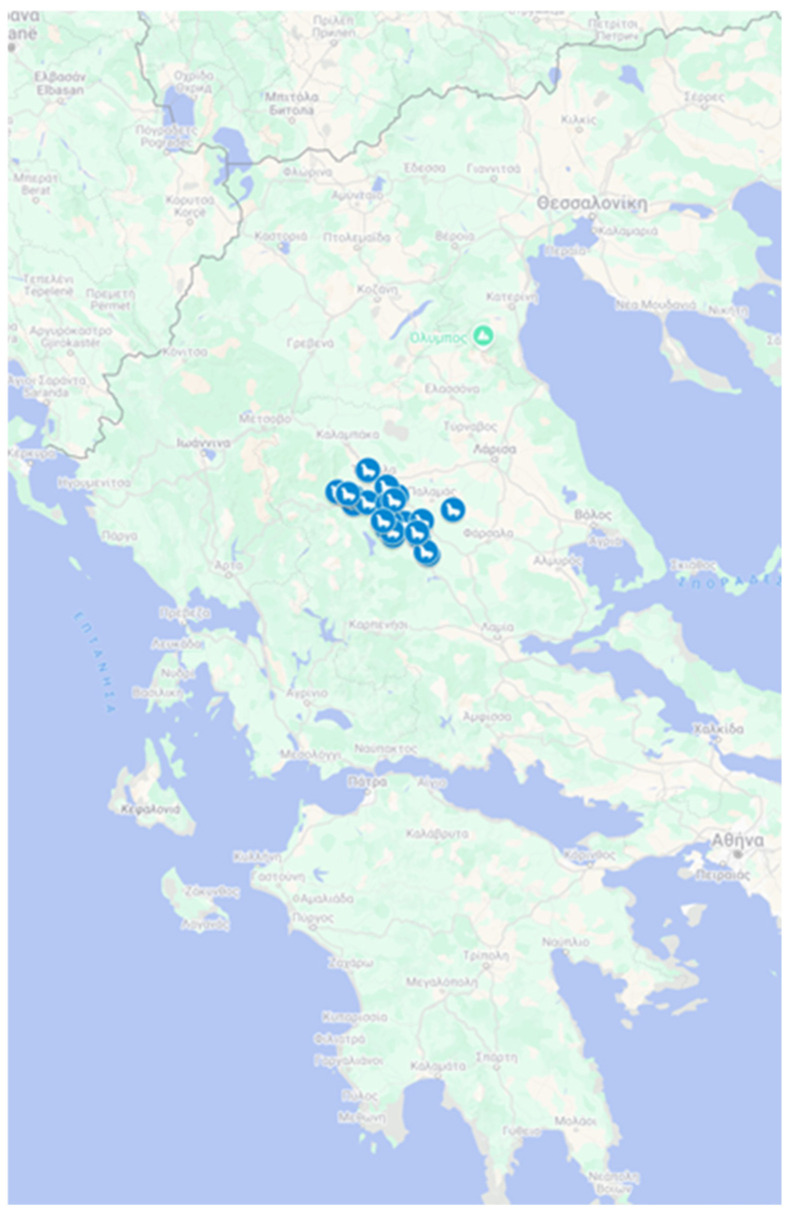
Farm mapping: the farms’ coordinates are represented with the blue dots in the map.

**Figure 2 animals-15-01104-f002:**
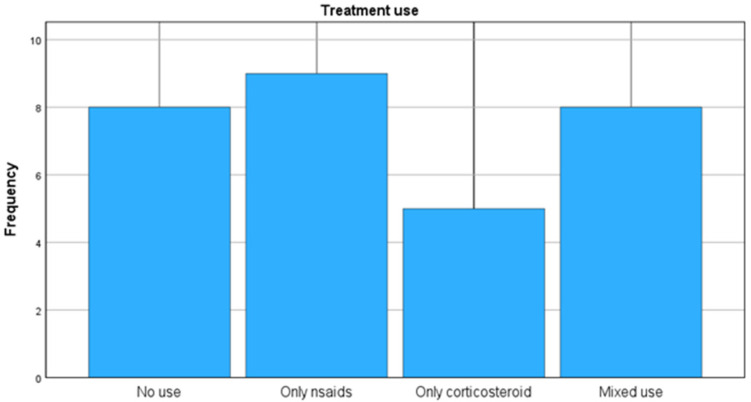
Distribution of treatment strategies among dairy sheep farms.

**Figure 3 animals-15-01104-f003:**
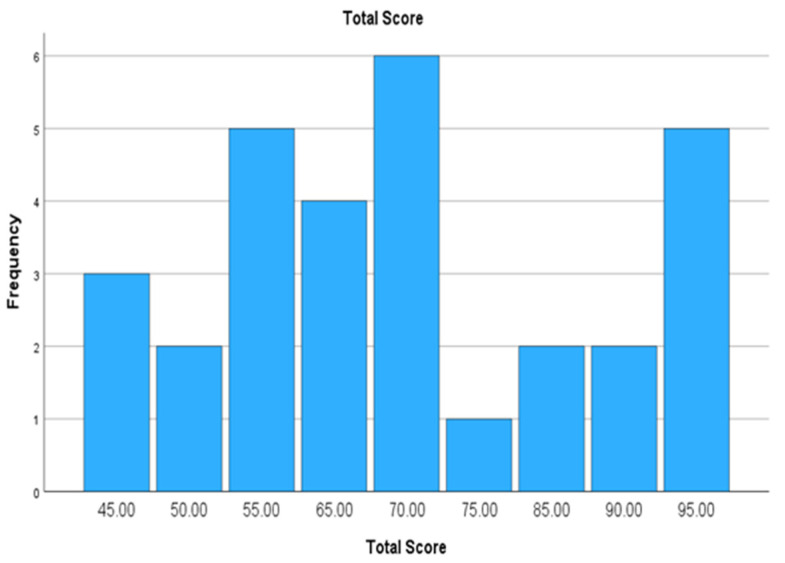
Distribution of Total Welfare Scores across farms.

**Figure 4 animals-15-01104-f004:**
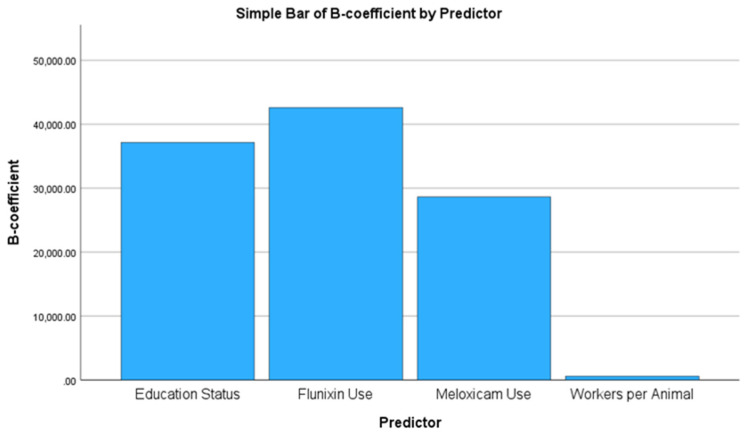
Bar chart of significant predictors of total milk yield.

**Table 1 animals-15-01104-t001:** Modified welfare scoring system for sheep based on Poulopoulou et al. [34].

Indicator	Normal Range/Requirements in Sheep	Score
Water supply	Adequate water suppliers (absence of waiting queue)	10
	Inadequate water suppliers (presence of waiting queue)	0
Space per animal	1.5 m^2^/ewe or more (1.2 m^2^ per ewe and 0.3 m^2^ per lamb)	10
	Less than 1.5 m^2^/ewe	0
BCS	BSC of 2–3 in lactating ewes and 3–4 in dry period ewes in 5% of the total herd, randomly picked ewes	10
	BSC of 2 in lactating ewes 2 and 3 in dry period ewes in 5% of the total herd, randomly picked ewes	5
	BSC of 1 in lactating ewes 1 and 1–2 in dry period ewes in 5% of the total herd, randomly picked ewes	0
Cleanliness	Clean in the upper body, belly, and udder in 5% of the total herd, randomly picked ewes	10
	Clean only in the upper body in 5% of the total herd, randomly picked ewes	0
Skin alterations	Absence of alterations in 5% of the total herd, randomly picked ewes	10
	Hairless spots in 5% of the total herd, randomly picked ewes	5
	Wounds in 5% of the total herd, randomly picked ewes	0
Avoidance behavior	5% of the total herd, randomly picked ewes, distanced less than 1 m	10
	5% of the total herd, randomly picked ewes, distanced more than 1 m	0
Hoof conformation	1 hoof trimming per year	10
	No hoof trimming in the previous year	0
Lameness	<5% presence in the flock	10
	5–10% presence in the flock	5
	>10% presence in the flock	0
Getting-up behavior	Physiological way of getting up of 5% of the total herd, randomly picked ewes	10
	Abnormal way of getting up of 5% of the total herd, randomly picked ewes	0
Dystocia occurrence	<5% in the flock	10
	5–10% in the flock	5
	>10% in the flock	0

**Table 2 animals-15-01104-t002:** Descriptive statistics of continuous variables in this study.

Variable	Min	Max	Mean	St. Deviation
Number of animals	45	640	175.3	134.40
Total milk yield (kg)	5394	236,739	50,589.0	49,946.4
Milk yield per sheep (kg)	119.9	383.8	261.26	67.63
Mean fat %	5.51	7.01	6.12	0.38
Mean proteins %	4.67	6.18	5.61	0.30
Flunixin meglumine vials	0.0	3.0	0.43	0.89
Meloxicam vials	0.0	13.0	1.43	3.17
Dexamethasone vials	0.00	5.00	0.87	1.36
Animals per worker (n)	30.0	200.0	88.06	39.3

**Table 3 animals-15-01104-t003:** Descriptive statistics for ordinal welfare and management variables.

Variable	Median	Mode	St. Deviation
Mean Body Condition Score	2.0	2.0	0.57
Housing Condition Score ^1^	2.0	2.0	0.73
Farmer’s age ^2^	2.0	2.0	0.70
Water supply	10.0	10.0	0.0
Space per animal	10.0	10.0	4.66
Body Condition Score	5.0	5.0	2.73
Cleanliness	10.0	10.0	4.66
Skin alterations	10.0	10.0	2.45
Avoidance behavior	0.0	0.0	4.90
Hoof conformation	2.5	0.0	5.0
Lameness	5.0	5.0	3.31
Getting-up behavior	10.0	10.0	0.0
Dystocia occurrence	10.0	10.0	2.52
Total score	70.0	70.0	16.92
Welfare score	2.0	2.0	0.81

^1^ 1 = Suboptimal, 2 = Moderate, and 3 = Optimal. ^2^ 1 < 35 years, 2 = 35–50 years, and 3 > 50 years.

**Table 4 animals-15-01104-t004:** Regression analysis of factors affecting total milk yield (kg).

Model	Unstandardized Coefficients		Standardized Coefficients	t	Sig.	Collinearity Statistics	
	B	Std. Error	Beta			Tolerance	VIF
(Constant)	−10,057.8	23,138.4		−0.435	0.668		
Mixed use	−30,338.4	21,444.9	−0.273	−1.415	0.171	0.244	4.1
Farmer’s education University degree	37,156.6	12,196.6	0.335	3.046	0.006	0.754	1.3
Workers per animal	610.0	152.4	0.480	4.003	<0.001	0.634	1.6
Flunixin use	42,616.3	14,077.1	0.367	3.027	0.006	0.619	1.6
Meloxicam use	28,623.0	13,621.7	0.291	2.101	0.047	0.473	2.1
Corticosteroid use	−3386.6	17,517.1	−0.034	−0.193	0.848	0.291	3.4
Farmer’s age	−7684.9	7978.9	−0.108	−0.963	0.346	0.723	1.3

Dependent variable: total milk yield (kg).

**Table 5 animals-15-01104-t005:** Model output: point Estimates and 95% confidence intervals (CIs) of model parameters.

Model	Dependent Variable	Factor	Point Estimate (95% CI)	*p*-Value
1	Flunixin meglumine	Intercept	0.02 (0.001–0.16)	-
		Herd Size	1.01 (1.005–1.025)	0.01
2	Meloxicam	Intercept	0.71 (0.11–4.6)	-
		Herd Size	0.95 (0.88–0.99)	0.03
		Milk yield herd (kg)	1.002 (1.001–1.004)	0.025
3	Dexamethasone	Intercept	0.47 (0.18–1.11)	-
		Farmers’ education	6.43 (1.15–52.25)	0.04
4	NSAID	Intercept	0 (0–0.0372)	-
		Milk yield herd (kg)	1.05 (1.018–1.11)	0.02
5	Welfare	Intercept	0 (0–0.0038)	-
		Milk yield herd (kg)	1.04 (1.02–1.08)	0.004

## Data Availability

Data supporting the conclusions of this study are available from the authors upon request.

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
