# Peer review of "Associations Between Non-Steroidal and Steroidal Anti-Inflammatory Drug Use, Welfare, and Milk Production in Dairy Sheep: A Multivariate Study"

_animals, 2025, doi:10.3390/ani15081104_

Round 1

Reviewer 1 Report

Comments and Suggestions for Authors

In the present study, the authors describe the association between non-steroidal and steroidal anti-inflammatory drug use in sheep. The study is well-designed, and the results are well-presented. However, the manuscript needs further amendments for publication.

General

Please proofread the whole manuscript and correct language errors

Line 30, for example, add the before swine

Abstract

Our findings indicate that 26.7% of sheep farmers did not utilize any analgesics, a figure that contrasts 30 sharply with high usage rates exceeding among farmers in swine and cattle sectors. Please add the percentage values for swine and cattle here (as added in line 406).

Introduction

Please modify the introduction make a strong case of the current knowledge, detail the gaps in our understanding and the explain how the present study fill those gaps in our understanding. The low levels of use of analgesics in sheep is mentioned in abstract, compared to cattle and swine. Please elaborate that in this section with appropriate references. Are there any specific reasons for reduced use of analgesics in sheep other than no availability of labelled drugs. Please provide more information with appropriate references.  

Materials and methods

The selection of 30 farms was based on both practical constraints (e.g., logistical and 128 financial limitations) and statistical considerations to ensure a representative sample. Still, the average herd size ranges from 45-700 animals. There may be wide variations among the use of analgesics and recording of the use of analgesics based on the size of the farm, as organised farms may have more facilities for the use of analgesics. All these factors need to be considered for the analysis. I would use block designing than a random design.

Figure1-please give a brief description of the figure

Please give an overview of the feeding practices followed on the farm. Please provide the veterinary service availability details on each farm.

Results

The data shows wide variation among farms. For example, milk production ranges from 5,394 kg to 236,739 kg, with a mean production of 50,589 kg 215 (SD = 49,946 kg). How did you account for these huge variations in the analysis.

Figure2 Please do not use capital letters in between the sentences.

Discussion

Line 406-Please check reference 8. This study is on sheep. This reference does not mention the use of analgesics in cattle.  

While NSAIDs facilitate recovery from illness and contribute to improved animal welfare [37], our results indicate that their use alone is not a reliable welfare indicator. Although ANOVA indicated a significant effect of treatment use on milk yield per sheep, post-hoc Bonferroni comparisons failed to detect statistically significant differences between specific treatment groups. This discrepancy may be attributed to the relatively small sample size, which could have limited the statistical power needed to identify pairwise differences. Alternatively, treatment effects on milk yield may not be sufficiently pronounced to generate clear distinctions between groups, possibly due to confounding management practices or variations in farm conditions. – I do not find these interpretations are conclusive. A pair-wise comparison with insufficient sample size may be avoided.

Future research with a larger and more diverse sample could offer a more comprehensive understanding of analgesic use patterns, associated welfare outcomes, and potential interventions to enhance pain management in sheep farming.- Is it a more diverse sample or a more uniform sample?

Comments on the Quality of English Language

Please proofread the whole manuscript

Author Response

Comments and Suggestions for Authors

In the present study, the authors describe the association between non-steroidal and steroidal anti-inflammatory drug use in sheep. The study is well-designed, and the results are well-presented. However, the manuscript needs further amendments for publication.

Thank you for your positive comment.

General

Please proofread the whole manuscript and correct language errors

Line 30, for example, add the before swine

Thank you for your comment. We revised the language errors, please check the manuscript. Please see the attachment

Abstract

Our findings indicate that 26.7% of sheep farmers did not utilize any analgesics, a figure that contrasts 30 sharply with high usage rates exceeding among farmers in swine and cattle sectors. Please add the percentage values for swine and cattle here (as added in line 406).

Thank you for your comment. The rates of analgesia use have been added. Please see the attachment Line 37

Introduction

Please modify the introduction make a strong case of the current knowledge, detail the gaps in our understanding and the explain how the present study fill those gaps in our understanding. The low levels of use of analgesics in sheep is mentioned in abstract, compared to cattle and swine. Please elaborate that in this section with appropriate references. Are there any specific reasons for reduced use of analgesics in sheep other than no availability of labelled drugs. Please provide more information with appropriate references.  

Thank you for your comment. We have improved the introduction, highlighted the under-utilization of analgesia in sheep. Please check L93-99.

Materials and methods

The selection of 30 farms was based on both practical constraints (e.g., logistical and 128 financial limitations) and statistical considerations to ensure a representative sample. Still, the average herd size ranges from 45-700 animals. There may be wide variations among the use of analgesics and recording of the use of analgesics based on the size of the farm, as organised farms may have more facilities for the use of analgesics. All these factors need to be considered for the analysis. I would use block designing than a random design.

Thank you for your insightful comment. We acknowledge that herd size and level of organization could influence analgesic use and record-keeping practices. However, our study aimed to capture the real-world variability of sheep farms in West Thessaly, and for this reason, we opted for simple random sampling from a stratified list of eligible farms, ensuring proportional representation of small, medium, and large operations. While block design may enhance internal validity in experimental studies, the observational nature of our research and the limited number of available farms (n=30) made simple random sampling more practical and representative of actual field conditions. To address potential confounding, we incorporated herd size and other key variables (e.g., education, housing) as covariates in multivariate models and used Propensity Score Matching (PSM) to control for baseline differences between treatment groups. This approach allowed us to retain ecological validity while minimizing bias and ensuring robust inference.

Figure1-please give a brief description of the figure

Thank you for your comment. You can find the description below figure 1, L185

Please give an overview of the feeding practices followed on the farm. Please provide the veterinary service availability details on each farm.

Thank you for your comment, you can find the feeding practices L173-178

Results

The data shows wide variation among farms. For example, milk production ranges from 5,394 kg to 236,739 kg, with a mean production of 50,589 kg 215 (SD = 49,946 kg). How did you account for these huge variations in the analysis.

Thank you for your insightful comment. The wide variation in total milk yield among the farms indeed reflects the inherent heterogeneity of the dairy sheep sector in West Thessaly. As farms were selected through random sampling, the final sample included both modernized, high-yielding operations (e.g., those using mechanized milking systems and formulated feeding) and more traditional, small-scale units still relying on manual milking and extensive management. This structural diversity is characteristic of the sector and enhances the generalizability of our findings.

Rather than restricting the sample to a homogeneous subset, we intentionally retained this variation to reflect real-world conditions. To ensure valid comparisons and minimize the impact of confounding, farm-level variables such as herd size, housing quality, labor intensity, and farmer education were included in our multivariate models. This approach allowed us to account for differences in productivity and management when assessing the association between analgesic use, welfare, and milk production

Figure2 Please do not use capital letters in between the sentences.

Thank you for your comment, please check the adjustments in Figure 2. Line 266

Discussion

Line 406-Please check reference 8. This study is on sheep. This reference does not mention the use of analgesics in cattle.  

Thank you for your comment, you can find the adjustment L458

While NSAIDs facilitate recovery from illness and contribute to improved animal welfare [37], our results indicate that their use alone is not a reliable welfare indicator. Although ANOVA indicated a significant effect of treatment use on milk yield per sheep, post-hoc Bonferroni comparisons failed to detect statistically significant differences between specific treatment groups. This discrepancy may be attributed to the relatively small sample size, which could have limited the statistical power needed to identify pairwise differences. Alternatively, treatment effects on milk yield may not be sufficiently pronounced to generate clear distinctions between groups, possibly due to confounding management practices or variations in farm conditions. – I do not find these interpretations are conclusive. A pair-wise comparison with insufficient sample size may be avoided.

Thank you for your comment. We agree that the limited sample size may constrain the statistical power of post-hoc pairwise comparisons, and we acknowledge that the Bonferroni correction is highly conservative in this context. Our intention was not to draw definitive conclusions based on pairwise differences, but rather to transparently report the outcome of the full ANOVA framework. To avoid overinterpretation, we revised the discussion to emphasize that the overall association between treatment use and milk yield was detected through ANOVA and regression modeling, while pairwise distinctions should be interpreted with caution given the sample size and potential confounding effects. See Lines 427 -435.

Future research with a larger and more diverse sample could offer a more comprehensive understanding of analgesic use patterns, associated welfare outcomes, and potential interventions to enhance pain management in sheep farming.- Is it a more diverse sample or a more uniform sample?

Thank you for your comment. In this case, by “more diverse sample” we refer to broader representation of farm types, management systems, and regions. Such diversity would allow for a more comprehensive understanding of real-world patterns in analgesic use. We agree that in experimental studies, a more uniform sample may be preferable to reduce variability; however, for observational research aiming to inform practical interventions across the sector, diversity enhances relevance and generalizability.

Reviewer 2 Report

Comments and Suggestions for Authors

The relative under-ultisation of analgesics in sheep is increasingly recognised. Reference 34 is very relevant and should be considered more as context in the introduction, as should these recent publications https://doi.org/10.3390/ani14070990 , https://www.mdpi.com/2076-2615/11/2/423 (the latter may also be useful to justify the grouping of MEL and FLU in this study)

The more novel aspect of this study is potential wider benefits beyond analgesia per se, where it is important to recognise that the authors are clear about the difference between association and causality. 

One aspect that should merit further discussion is summarised Line 376-381; “In this study, higher welfare status was associated with greater milk production, yet NSAID administration did not correlate with improved welfare scores. This finding may be attributed to poor welfare conditions increasing disease incidence, thereby leading to a greater need for analgesic intervention. While NSAIDs facilitate recovery from illness and contribute to improved animal welfare [37], our results indicate that their use alone is not a reliable welfare indicator. , coupled with Line 454-457 “While NSAID administration (e.g., flunixin, meloxicam) was linked to higher milk yields, its lack of correlation with improved welfare scores suggests that poor welfare conditions may increase disease prevalence, necessitating greater reliance on analgesics.” This could suggest ( I recognise that this in not what the authors are saying) that analgesics are mainly a ‘rescue’ tool for poor welfare, and less important as a tool in high welfare production. Whilst they may be required less, there may well be indication for their use on all farms.

Author Response

We thank the reviewers for their thorough and constructive feedback, which greatly contributed to improving the quality and clarity of our manuscript. We have carefully considered all comments and revised the manuscript accordingly. Below, we provide a detailed point-by-point response to each suggestion and explain the corresponding changes made in the text. Line numbers refer to the revised version of the manuscript. We trust that the revised version addresses all concerns and is now suitable for publication.

The relative under-ultisation of analgesics in sheep is increasingly recognised. Reference 34 is very relevant and should be considered more as context in the introduction, as should these recent publications https://doi.org/10.3390/ani14070990 , https://www.mdpi.com/2076-2615/11/2/423 (the latter may also be useful to justify the grouping of MEL and FLU in this study)

Thank you for your comment. We have improved the introduction, highlighted the under-utilization of analgesia in sheep. Please check the revised manuscript

The more novel aspect of this study is potential wider benefits beyond analgesia per se, where it is important to recognise that the authors are clear about the difference between association and causality. 

Thank you for your positive comment.

One aspect that should merit further discussion is summarised Line 376-381; “In this study, higher welfare status was associated with greater milk production, yet NSAID administration did not correlate with improved welfare scores. This finding may be attributed to poor welfare conditions increasing disease incidence, thereby leading to a greater need for analgesic intervention. While NSAIDs facilitate recovery from illness and contribute to improved animal welfare [37], our results indicate that their use alone is not a reliable welfare indicator. , coupled with Line 454-457 “While NSAID administration (e.g., flunixin, meloxicam) was linked to higher milk yields, its lack of correlation with improved welfare scores suggests that poor welfare conditions may increase disease prevalence, necessitating greater reliance on analgesics.” This could suggest ( I recognise that this in not what the authors are saying) that analgesics are mainly a ‘rescue’ tool for poor welfare, and less important as a tool in high welfare production. Whilst they may be required less, there may well be indication for their use on all farms.

Thank you for your comment. We acknowledge that the interpretation of Lines 376–381 and 454–457 could unintentionally suggest that analgesics are primarily used as a rescue measure in low-welfare farms and less relevant in high-welfare settings. To clarify this important point, we have revised the discussion to emphasize that while analgesic use may be more frequent in farms with poorer welfare due to increased disease incidence, this does not diminish the relevance of pain management in farms with high welfare standards. Painful conditions can still occur even under optimal management—for example, following routine procedures or accidental injury—and therefore, appropriate analgesic use remains essential across all farming systems. We have inserted a clarifying paragraph in the Discussion section (Line 420), immediately after discussing the lack of association between NSAID use and welfare scores.

Round 2

Reviewer 1 Report

Comments and Suggestions for Authors

The authors have made necessary corrections in the manuscript and hence can be accepted.